# GPD-VVTO: Preserving Garment Details in Video Virtual Try-On

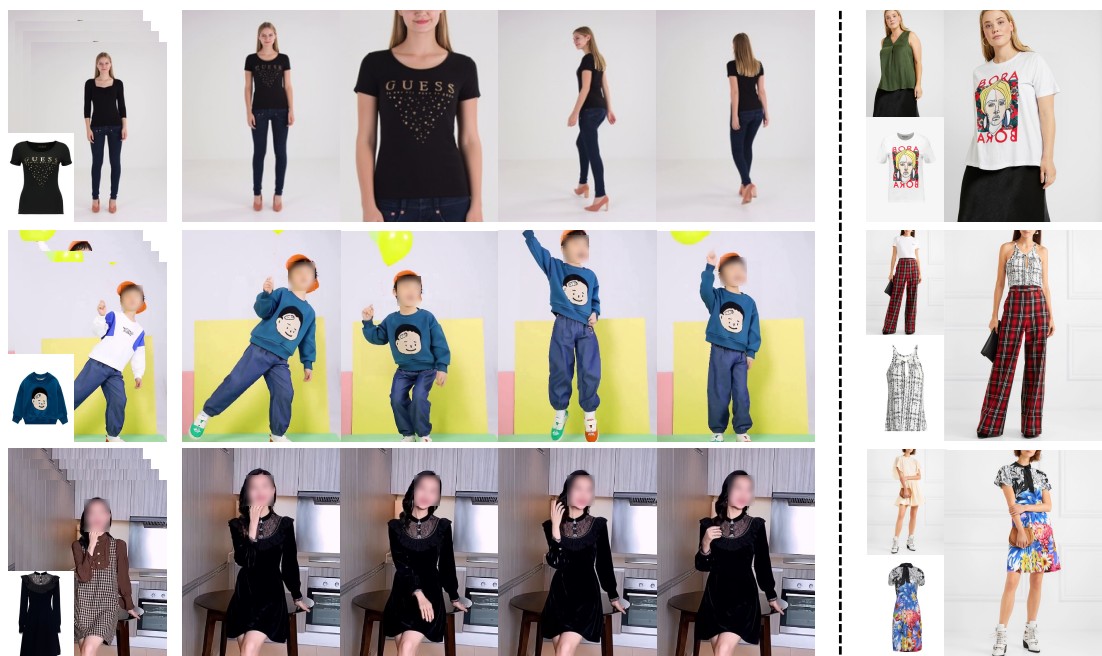
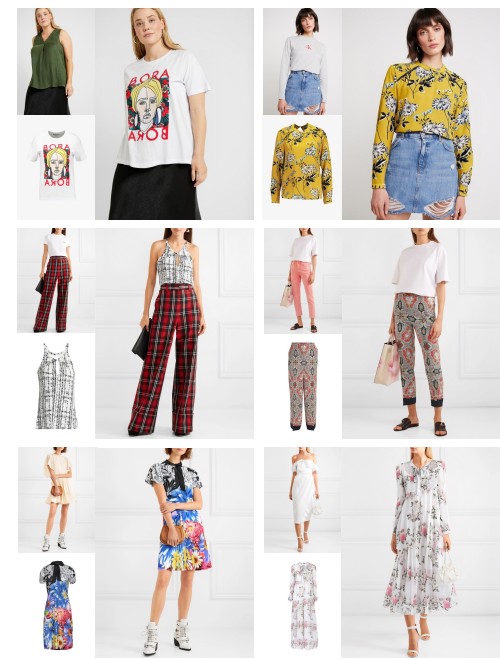

**Figure 1:** *Left:* video virtual try-on results (512 × 384) generated by our method on VVT (1-st row) and our (2-nd and 3-rd row) datasets. The faces are blurred due to privacy concerns. *Right:* image-based virtual try-on results (1024 × 768) generated by our method on VITON-HD (1-st row) and Dress Code (2-nd and 3-rd row) datasets. Best viewed on a zoomed, color monitor.

## ABSTRACT

Video Virtual Try-On aims to transfer a garment onto a person in the video. Previous methods typically focus on image-based virtual try-on, but directly applying these methods to videos often leads to temporal discontinuity due to inconsistencies between frames. Limited attempts in video virtual try-on also suffer from unrealistic results and poor generalization ability. In light of previous research, we posit that the task of video virtual try-on can be decomposed into two key aspects: (1) single-frame results are realistic and natural, while retaining consistency with the garment; (2) the person's actions and the garment are coherent throughout the entire video. To address these two aspects, we propose a novel two-stage framework based on Latent Diffusion Model, namely Garment-Preserving Diffusion for Video Virtual Try-On (GPD-VVTO). In the first stage, the model is trained on single-frame data to improve the ability of generating high-quality try-on images. We integrate both low-level texture features and high-level semantic features of the garment into the denoising network to preserve garment details while ensuring a natural fit between the garment and the person. In the second stage, the model is trained on video data to enhance temporal consistency. We devise a novel Garment-aware Temporal Attention (GTA) module that incorporates garment features into temporal attention, enabling the model to maintain the fidelity to the garment during temporal modeling. Furthermore, we collect a video virtual try-on dataset containing high-resolution videos from diverse scenes, addressing the limited variety of current datasets in terms of video background and human actions. Extensive experiments demonstrate that our method outperforms existing state-of-the-art methods in both image-based and video-based virtual try-on tasks, indicating the effectiveness of our proposed framework.

## CCS CONCEPTS

• **Computing methodologies** → **Computer vision**.

## KEYWORDS

Virtual Try-On, Diffusion Models, Video Editing

## 1 INTRODUCTION

Given a segment of human video and an image of a garment, Video Virtual Try-On aims to synthesize a natural and coherent video of

the specific person wearing the provided garment. With the rise of e-commerce, this technology has garnered widespread attention in the industry, offering consumers an immersive and interactive online shopping experience.

Previous virtual try-on methods [6, 12, 15, 23, 26, 28, 34, 42, 43, 52] typically focus on image-based operations, employing Generative Adversarial Networks [11] (GANs) or Diffusion Models [17] as the foundational architecture. For instance, GP-VTON [42] warps local garment parts individually and assembles them via global parsing. StableVITON [23] introduces zero cross-attention modules to learn the semantic correspondence between the human and the garment. However, these methods still struggle to balance the naturalness of person-garment blending and the fidelity to the garment's appearance. Moreover, the experience provided by image-based virtual try-on is far less immersive for consumers compared to video virtual try-on. Even though we can directly apply these image-based methods to video virtual try-on with a frame-by-frame method, this often leads to discontinuity in the generated videos due to inconsistencies between frames.

Several efforts have been made in the domain of video virtual try-on. FW-GAN [8] integrates optical flow prediction to warp the preceding frames for subsequent frame generation. MV-TON [51] introduces a memory refinement module that aggregates spatiotemporal information of multiple frames to enhance the generated frames. ClothFormer [21] proposes a novel warping and tracking module to ensure temporal consistency in garment warping. Despite the improvements in the spatiotemporal consistency of generated videos, these methods rely on GAN-based frameworks and necessitate separate warping modules, resulting in poor realism of generated videos and limited generalization of the models.

In light of previous research, we posit that the task of video virtual try-on can be decomposed into two key aspects. First, every single frame in the generated video is realistic and natural, while keeping the background regions unchanged and preserving the details of the garment as much as possible. Second, the actions of the person and the appearance of the garment are consistent and coherent throughout the entire video.

To address the aforementioned two aspects, we propose a novel two-stage video virtual try-on method based on Latent Diffusion Model [32], namely **G**arment-**P**reserving **D**iffusion for **V**ideo **V**irtual **T**ry-**O**n (GPD-VVTO). *In the first stage,* the model is trained on image-based virtual try-on task to ensure the high quality of individual frames. To retain more garment details, we extract both high-level semantic features and low-level texture features of the garment and inject them into the denoising network. On the one hand, global semantic features of the garment at different abstraction levels are extracted by a pre-trained DINOv2 [30] encoder, which are utilized to conduct cross-attention with feature maps of different spatial resolution in our proposed Semantic-enhanced Cross-Attention (SCA) modules. On the other hand, we employ a U-Net-based garment encoder to extract multi-level dense features of the garment. The obtained garment features are incorporated into the main U-Net within our proposed Joint Spatial Attention (JSA) modules and jointly perform self-attention with the features of the main U-Net. Through the combination of global and local garment information, the model is capable of generating realistic try-on images while preserving the overall style and fine-grained

details of the garment. Furthermore, the JSA module implicitly establishes a soft spatial correspondence between the human and the garment, facilitating their natural blending. *In the second stage,* the model is fine-tuned on video data to enhance the temporal consistency of the generated videos with additional temporal modules. To maintain the fidelity to the garment during temporal modeling, we devise a novel Garment-aware Temporal Attention (GTA) module, which leverages features extracted from the target garment to perform temporal attention. Specifically, we introduce an extra spatial cross-attention operation that shares weights with the JSA module, while the human video features serve as queries, and the garment features serve as keys and values. Therefore, the person-garment correspondence established by the JSA module enables the resulting feature maps to represent the garment features spatially aligned to the human video. Subsequently, we append these features to the feature sequence of the human video in the temporal dimension for temporal self-attention. In this way, the model can not only improve the continuity of the generated videos by referencing features from preceding and subsequent frames, but also preserve more garment details by leveraging features from the garment encoder.

In addition, current video virtual try-on datasets [8] lack diversity in terms of video backgrounds and human actions, diverging significantly from real-world scenarios. To address the limitation of data, we collect a high-resolution video virtual try-on dataset from well-known e-commerce platforms, which contains videos of numerous models performing various actions in diverse scenes and the corresponding garment images. Extensive experiments demonstrate that our method outperforms previous state-of-the-art methods on both public and our collected datasets. We also evaluate our model of the first stage on two image-based virtual try-on benchmarks [5, 29]. Our model surpasses existing image-based virtual try-on methods in all metrics, further validating the effectiveness of our proposed modules.

In summary, our contributions are as follows:

- We explore the capability of diffusion models in video virtual try-on for the first time, and propose a two-stage framework that leverages both global semantic features and local texture features of the garment to preserve garment details.
- We devise a novel Garment-aware Temporal Attention module that integrates features of the garment into temporal attention, improving the temporal consistency of the generated videos while maintaining the fidelity to the garment.
- We collect a video virtual try-on dataset with more diversity to resolve the limited variety of current datasets in terms of video background and human actions. Extensive experiments demonstrate that our method outperforms previous state-of-the-art methods on both image-based and video-based virtual try-on datasets.

## 2 RELATED WORK

### 2.1 Image-based Virtual Try-On

**GAN-based methods.** Most GAN-based methods [15, 26, 34, 42] follow a two-stage generation framework, first deforming the target garment to fit the person's body, and then fusing the deformed garment with the reference person. HR-VITON [26] proposes a try-on condition generator that predicts simultaneously the warped

garment and the segmentation map, removing the misalignment and handling occlusions of clothes by body parts naturally. GP-VTON [42] employs local flows to warp garment parts individually and assembles the local warped results via global garment parsing, avoiding the garment squeezing and texture distortion problems during warping. However, the generational capacity of GANs significantly restricts the performance of GAN-based methods.

**Diffusion-based methods.** Recently, diffusion models have emerged as a promising alternative to GANs due to their exceptional performance in generating high-quality images at high resolutions. Consequently, there has been growing interest in exploring the application of diffusion models in virtual try-on tasks [6, 12, 23, 28, 43, 52]. TryOnDiffusion [52] introduces a try-on framework with parallel U-Nets to handle garment warping and person blending simultaneously. Despite its promising performance, this approach is computationally intensive due to the requirement for multiple diffusion models in concatenation. DCI-VTON [12] and LaDI-VTON [28] treat virtual try-on as an exemplar-based inpainting task, and fine-tunes the pre-trained inpainting diffusion model on virtual try-on datasets. StableVITON [23] eliminates the warping module and employs zero cross-attention blocks to learn semantic correspondence between person and garment in an end-to-end manner. However, it is still difficult for these methods to preserve detailed textures of the garment. Our method employs a pre-trained DINOv2 [30] encoder to provide global semantic features and a parallel U-Net-based garment encoder to extract local texture features of the garment, which are integrated into the SCA and JSA modules of the main U-Net, leading to a better preservation of garment details.

## 2.2 Video Virtual Try-On

Compared with image virtual try-on, video virtual try-on [8, 21, 51] offers a more user-friendly and natural try-on experience. FW-GAN [8] enhances spatiotemporal smoothing in generated videos by incorporating optical flow with a warping net for person and garment manipulation. MV-TON [51] proposes a memory refinement operation to improve the details of initially generated results by referring to the previous frames. ClothFormer [21] adopts Transformer [39] architecture and proposes a new warping module and tracking module to warp the garment with temporal consistency. However, previous methods commonly utilize GAN-based frameworks and require separate warping modules, leading to a complex processing workflow. In this work, we explore the capability of diffusion models in video virtual try-on. We forgo the standalone warping module and enable the model to implicitly learn fine-grained spatial correspondence between the person and garment within the JSA module.

## 2.3 Diffusion Models for Video Generation

Given the notable advancements of diffusion models in image generation [27, 31–33, 37, 45, 48], numerous studies have extended text-to-image (T2I) diffusion models to video-related tasks, including text-to-video (T2V) [2, 7, 9, 14], image-to-video (I2V) [20, 44, 49] and video editing [4, 41, 53]. AnimateDiff [14] introduces a plug-and-play motion module that can be seamlessly integrated into personalized T2I models, forming an animation generator. Pixel-Dance [49] proposes straightforward modifications on the T2V U-Net, incorporating the latent of image condition with input noise to facilitate I2V generation. Tune-A-Video [41] enhances self-attention layers to enable each frame to attend to both the previous and first frames. Cut-and-Paste [53] integrates an extra reference image with the text condition to achieve more precise and fine-grained video editing. As a specific case of video editing, video virtual try-on necessitates maintaining the temporal consistency of the video while ensuring the fidelity to the garment. Our proposed GTA module incorporates features of the garments into temporal attention mechanisms, enabling the model to simultaneously address temporal continuity and garment consistency.

## 3 METHOD

### 3.1 Preliminary: Stable Diffusion

Our proposed GPD-VVTO builds upon Stable Diffusion (SD), an advanced image generation model derived from Latent Diffusion Model (LDM) [32]. LDM operates the diffusion process in the latent space to mitigate computational complexity.

During training, a latent encoder [25, 38] compresses the input image $\mathbf{x}$ into a latent code $\mathbf{z_0} = \mathcal{E}(\mathbf{x})$ using an encoder $\mathcal{E}$. SD performs forward diffusion process by adding Gaussian noise to the latent:

$$q(\mathbf{z}_t|\mathbf{z_0}) = \mathcal{N}(\mathbf{z}_t; \sqrt{\alpha_t}\mathbf{z_0}, \sqrt{1-\alpha_t}\mathbf{I}), \quad (1)$$

where $t$ stands for the number of diffusion timesteps and $\{\alpha_i\}_{i=1}^t$ control the diffusion schedule. The denoising U-Net is trained to reverse this process by predicting the added noise. The optimization objective of this process is expressed as:

$$\mathcal{L} = \mathbb{E}_{\mathbf{z}_t,c,\epsilon \sim \mathcal{N}(0,1),t}||\epsilon - \epsilon_\theta(\mathbf{z}_t, c, t)||_2^2, \quad (2)$$

where $\epsilon_\theta$ represents the function of U-Net, $t$ stands for the number of diffusion timesteps and $c$ is the embedding of conditional information. In the original SD, conditional text is embedded using a CLIP-based transformer. Each typical block in denoising U-Net contains three types of computation: 2D convolution, self-attention and cross-attention with conditional text embedding.

During inference, $\mathbf{z}_t$ is randomly sampled from a Gaussian distribution, and progressively denoised to obtain $\mathbf{z_0}$ following a pre-defined sampling schedule [17, 36]. Finally, a latent decoder $\mathcal{D}$ decodes the result back into the image space.

### 3.2 Overview

An overview of GPD-VVTO is illustrated in Figure 2 (a). Given a segment of human video $\{\mathbf{x}\}_1^T = \{\mathbf{x}_1, ..., \mathbf{x}_T\} \in \mathbb{R}^{T \times 3 \times H \times W}$ and a garment image $\mathbf{c} \in \mathbb{R}^{3 \times H \times W}$, where $H$ and $W$ denote the height and width of each frame and $T$ denotes the length of the segment, GPD-VVTO aims to synthesis a realistic and natural video sequence $\{\bar{\mathbf{x}}\}_1^T \in \mathbb{R}^{T \times 3 \times H \times W}$ that presents the person in $\{\mathbf{x}\}_1^T$ wearing the garment $\mathbf{c}$.

GPD-VVTO contains two U-Net branches. On the one hand, the person video $\{\mathbf{x}\}_1^T$ and the cloth-agnostic video $\{\mathbf{a}\}_1^T$ are encoded to the latent space by the encoder of VAE $\mathcal{E}$. A Gaussian noise is then added to $\mathcal{E}(\{\mathbf{x}\}_1^T)$ to form the noisy latent $\{\mathbf{z}\}_1^T \in \mathbb{R}^{T \times 4 \times h \times w}$, where $h = H/8$ and $w = W/8$. The main U-Net is an inpainting network that takes a 9-channel tensor as input, with 4 channels of

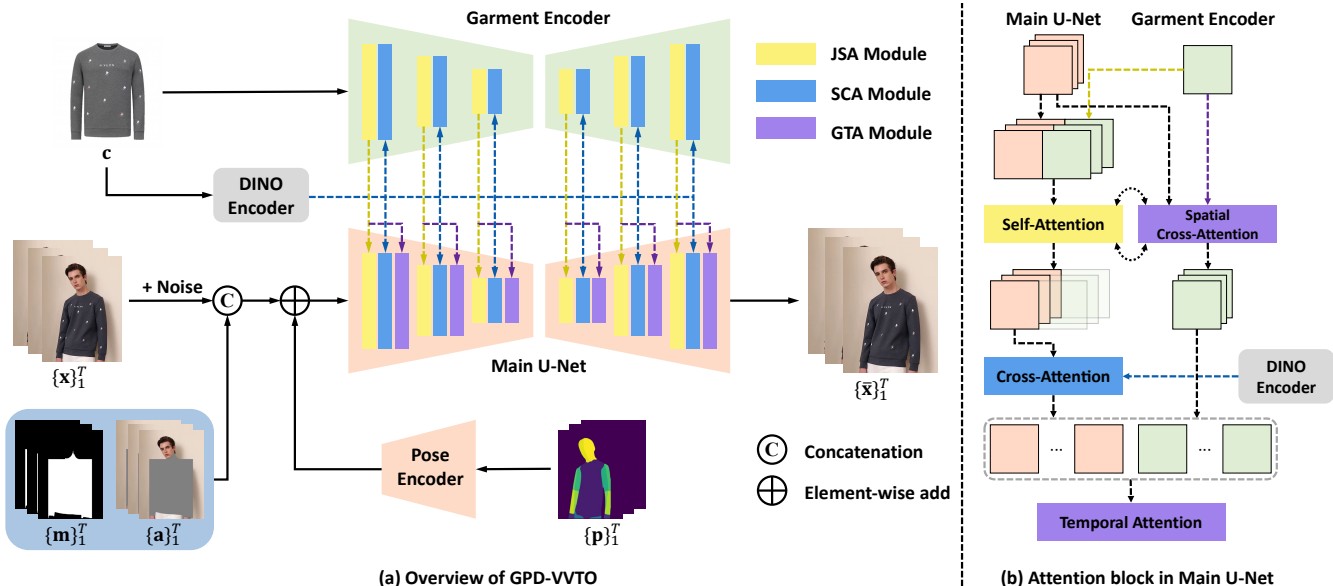

**(a) Overview of GPD-VVTO**

**(b) Attention block in Main U-Net**

Figure 2: (a) Overall architecture of GPD-VVTO. The main U-Net takes the concatenation of the noisy latent of the video, the latent of the cloth-agnostic video and the binary agnostic mask sequence as input. DensePose sequence is encoded by the pose encoder and added to the input of the main U-Net. The garment image is processed by the U-Net-based garment encoder and the DINO encoder to extract local texture and global semantic features respectively. The garment features are injected into the main U-Net within the JSA, SCA and GTA modules to preserve garment details. The encoder and decoder of VAE are not shown for clarity. (b) Detailed illustration of the attention block in the main U-Net. *Left branch*: the feature maps from the main U-Net and garment encoder are concatenated in the spatial dimension to jointly perform self-attention (JSA module). Only the left half (that from the main U-Net) of the obtained feature map is utilized to conduct cross-attention (SCA module) with features extracted by the DINO encoder. *Right branch* (GTA module): we compute a spatial cross-attention that shares weights with the self-attention operation, where the features from main U-Net serve as queries and garment features serve as keys and values. The obtained feature sequence contains garment features spatially aligned with the human video, which is appended to the human feature sequence from the main U-Net to perform temporal attention. Best viewed in color.

the noisy latent of the video, 4 channels of the latent of the cloth-agnostic video and 1 channel of the binary agnostic mask sequence. On the other hand, the garment encoder is a standard text-to-image diffusion model with the latent of the garment $\mathcal{E}(\mathbf{c}) \in \mathbb{R}^{4 \times h \times w}$ as input. Both the main U-Net and the garment encoder receive the embedding of garment image extracted by the DINO encoder for cross-attention. To further enhance the preservation of the body shape and pose of the person, the DensePose sequence $\{\mathbf{p}\}_1^T$ is embedded by a zero-initialized lightweight fully convolutional pose encoder and incorporated into the main U-Net via element-wise addition.

### 3.3 Pre-processing

To construct the cloth-agnostic video $\{\mathbf{a}\}_1^T$ and agnostic mask sequence $\{\mathbf{m}\}_1^T$, we first extract the human parsing results using Graphonomy [10] in a frame-by-frame manner, and mask out regions related to the garment (*e.g.*, clothes, coat, and body skin for upper-body garments). Subsequently, we utilize DWPose [47] results to mask the arms, leaving the hands exposed for better realism of the generation results. Finally, we take the bounding rectangle of the mask to eliminate the influence of the original garment style on the results. Areas unrelated to clothing are left exposed to better

preserve the identity of the person. We utilize [13] to extract the DensePose sequence frame by frame as the input of pose encoder.

### 3.4 Semantic-enhanced Cross-Attention

In the original SD, cross-attention is performed using text embeddings extracted by CLIP text encoder. To better adapt to the virtual try-on task, we replace these embeddings with features extracted from the garment image by DINOv2 [30], which contain rich global semantic information (*e.g.*, style). Previous works [23, 46] only utilize the last hidden state for all cross-attention modules. However, since different blocks in the U-Net have varying degrees of spatial dimension compression, semantic features of different abstraction levels are required.

To make full use of semantic features of different abstraction levels, we propose a Semantic-enhanced Cross-Attention (SCA) module that applies multiple intermediate features from the DINO encoder to different blocks of the two U-Net branches. Specifically, we define a mapping function $\sigma : \{1, ..., N\} \rightarrow \{1, ..., M\}$, where $N$ denotes the number of cross-attention modules in the U-Net, $M$ denotes the number of layers in DINO encoder, and $\sigma(i)$ represents which layer of semantic features should be utilized in the $i$-th cross-attention module. The operations in the SCA module can be

expressed as follow:

$$\tilde{\mathbf{f}}^i = \text{Attn}(\phi_q^i(\hat{\mathbf{f}}^i), \phi_k^i(\mathbf{s}^{\sigma(i)}), \phi_v^i(\mathbf{s}^{\sigma(i)})), \tag{3}$$

where $\hat{\mathbf{f}}^i$ stands for the feature map before the $i$-th cross-attention module in the U-Net, $\mathbf{s}$ denotes the semantic features extracted by DINO encoder, and $\phi_q^i, \phi_k^i, \phi_v^i$ are linear projection layers for queries, keys and values respectively. By incorporating semantic features from different abstraction levels into corresponding blocks, the model can better preserve the overall style of the garment.

## 3.5 Joint Self-Attention

Although injecting global semantic information of the garment helps preserve the overall style, the features extracted by the DINO encoder do not attach importance to local texture features, resulting in the loss of many local details (*e.g.*, text, lines and patterns). Hence, we leverage a parallel garment encoder to extract multi-level dense features of the garment, which are integrated with the features from main U-Net to perform self-attention in the Joint Self-Attention (JSA) modules. To ensure compatibility between garment features and human features, the garment encoder adopts the same architecture as the main U-Net, but is initialized using the weights of pre-trained text-to-image Stable Diffusion model. In each JSA module, the human feature map from the main U-Net $\mathbf{f}^i$ is concatenated with the corresponding garment feature map from the garment encoder $\mathbf{g}^i$ to jointly compute self-attention. Concretely, the $i$-th JSA module in the main U-Net is computed as follow:

$$\hat{\mathbf{h}}^i = \text{Attn}(\psi_q^i(\mathbf{h}^i), \psi_k^i(\mathbf{h}^i), \psi_v^i(\mathbf{h}^i)), \tag{4}$$

where $\mathbf{h}^i = \text{Concat}(\mathbf{f}^i, \mathbf{g}^i)$ is the concatenation of human feature map and garment feature map in the spatial dimension, and $\psi_q^i$, $\psi_k^i, \psi_v^i$ are linear projections. Afterwards, only the half from the main U-Net undergoes further computation. By integrating the dense garment feature maps abound of local texture information into self-attention modules, the main U-Net can selectively extract local features from the garment feature map, achieving the effective transfer of detailed garment appearance. Furthermore, through the joint self-attention of human and garment features, we establish a soft spatial correspondence between the human image and the garment image, implicitly deforming the garment to fit the pose of the person more naturally.

## 3.6 Garment-aware Temporal Attention

Both the SCA and JSA modules focus on the realism and fidelity of individual frame results without establishing correlations between frames, thus unable to address the issue of temporal inconsistency. To efficiently transform an image generation model into a video generation model, AnimateDiff [14] proposes a plug-and-play motion module that involves temporal attention to improve the smoothness of generated videos. However, since other parameters of the model are not updated during the training of motion modules, there is a stagnation or even degradation in the quality of single-frame results. Moreover, fine-tuning the entire model on video data requires abundant computational resources and time.

To overcome this issue, we devise a novel Garment-aware Temporal Attention (GTA) based on the motion module, which preserves fidelity to the garment by referencing garment features during the

temporal modeling process. As illustrated in the right branch of Figure 2 (b), we conduct an extra spatial cross-attention in parallel with the JSA module, with the human features from the main U-Net as queries and the garment features from the garment encoder as keys and values:

$$\hat{\mathbf{g}}^i = \text{Attn}(\psi_q^i(\mathbf{f}^i), \psi_k^i(\mathbf{g}^i), \psi_v^i(\mathbf{g}^i)). \tag{5}$$

It is worth noting that the feature maps used in this operation (*i.e.*, $\mathbf{f}^i$ and $\mathbf{g}^i$), as well as the weights of the linear projection layers (*i.e.*, $\psi_q^i, \psi_k^i$ and $\psi_v^i$), are the same as those in the JSA module. Therefore, the learned spatial correspondence between the person and the garment in the JSA module remains effective. In this way, the obtained feature map $\hat{\mathbf{g}}^i$ represents the garment features corresponding to each spatial position in the human image, and it is spatially aligned with the human feature map $\mathbf{f}^i$. Subsequently, the human feature maps $\{\tilde{\mathbf{f}}^i\}_1^T$ are concatenated with $\{\hat{\mathbf{g}}^i\}_1^T$ in the temporal dimension to jointly compute temporal self-attention. Similar to the JSA module, only the sequence from the main U-Net is retained for further operations. By incorporating aligned garment features into the temporal attention, we simultaneously enhance the temporal continuity of the video and the quality of individual frames.

## 3.7 Training and Inference

**Training scheme.** The training scheme of GPD-VVTO comprises two stages:

In the first stage, we take single frames as input and initialize the main U-Net as a 2D inpainting model, improving the model's ability of generating high-quality try-on images. All parameters except for the pre-trained DINO encoder are updated. To further align basic features of the results, we compute perceptual loss [22] utilizing a pre-trained VGG [35] model. The total optimization objective of the first stage is expressed as follow:

$$\mathcal{L}_1 = \mathbb{E}_{\hat{\mathbf{z}}|_t, \mathbf{p}, \mathbf{c}, \epsilon, t} ||\epsilon - \epsilon_\theta(\hat{\mathbf{z}}|_t, \mathbf{p}, \mathbf{c}, t)||_2^2 + \lambda \mathcal{L}_{perceptual}, \tag{6}$$

where $\hat{\mathbf{z}}|_t = \text{Concat}(\mathbf{z}|_t, \mathbf{m}, \mathbf{a})$ is the input of the main U-Net at timestep $t$ and $\lambda$ is the weight of the perceptual loss.

In the second stage, we fine-tune the model on video data by inflating the 2D convolutions to spatial-only pseudo-3D convolutions [14, 19], and insert GTA modules to enhance the temporal consistency. We only update the parameters of GTA modules and linear layers that project the features extracted by DINO encoder to the same dimensions as the U-Nets. The total optimization objective of the second stage is expressed as follow:

$$\mathcal{L}_2 = \mathbb{E}_{\{\hat{\mathbf{z}}\}_1^T|_t, \{\mathbf{p}\}_1^T, \mathbf{c}, \epsilon, t} ||\epsilon - \epsilon_\theta(\{\hat{\mathbf{z}}\}_1^T|_t, \{\mathbf{p}\}_1^T, \mathbf{c}, t)||_2^2. \tag{7}$$

**Classifier-free guidance.** We apply classifier-free guidance [18] on the garment image during inference to obtain more robust results. Specifically, we randomly set $\mathbf{c} = \emptyset$ during the whole training phase, with all elements in $\emptyset \in \mathbb{R}^{3 \times H \times W}$ equal to zero. In this way, the model is trained on both conditional and unconditional settings. During inference, we utilize a scalar guidance scale $s_g \geq 1$ to combine the conditional and unconditional results. For instance, the final noise predicted by the model in the first stage is as below:

$$\hat{\epsilon}_\theta(\hat{\mathbf{z}}|_t, \mathbf{p}, \mathbf{c}, t) = s_g \cdot \epsilon_\theta(\hat{\mathbf{z}}|_t, \mathbf{p}, \mathbf{c}, t) + (1 - s_g) \cdot \epsilon_\theta(\hat{\mathbf{z}}|_t, \mathbf{p}, \emptyset, t). \tag{8}$$

**Inference with sliding windows.** To enhance the temporal coherence and smoothness for long videos, we follow [44] to adopt

**Table 1: Quantitative results of video virtual try-on task on the VVT and our dataset. GPD-VVTO† denotes the first-stage model of our method. StableVITON‡ denotes the adaptation of StableVITON to Video Virtual Try-On task by inserting temporal attention modules. * means the results are from previous works.**

| Category | Method | VVT | | | Ours | | |
|---|---|---|---|---|---|---|---|
| | | SSIM ↑ | LPIPS ↓ | VFID ↓ | SSIM ↑ | LPIPS ↓ | VFID ↓ |
| Image-based VTON | LaDI-VTON [28] | 0.878 | 0.190 | 5.88 | 0.644 | 0.213 | 10.88 |
| | StableVITON [23] | 0.902 | 0.078 | 3.54 | 0.697 | 0.189 | 7.54 |
| | **GPD-VVTO† (Ours)** | 0.922 | 0.058 | 1.79 | 0.745 | 0.161 | 4.82 |
| Video VTON | MV-TON* [51] | 0.853 | 0.233 | 8.37 | - | - | - |
| | ClothFormer* [21] | 0.921 | 0.081 | 3.97 | - | - | - |
| | StableVITON‡ [23] | 0.903 | 0.080 | 4.05 | 0.701 | 0.190 | 6.52 |
| | **GPD-VVTO (Ours)** | **0.928** | **0.056** | **1.28** | **0.760** | **0.160** | **3.98** |

a sliding window method during inference. We divide the long video into multiple overlapping segments of length $T$ and perform inference on each segment. For overlapping frames, the final result is obtained by taking the average over each inference.

## 4 EXPERIMENTS

### 4.1 Datasets

**Video Virtual Try-On.** We conduct the experiments of video virtual try-on using VVT dataset [8] and our collected dataset. The VVT dataset contains 791 video clips with a resolution of $256 \times 192$. Following previous works [8, 21], we partition it into a training set of 661 clips and a test set of 130 clips. However, the VVT dataset lacks diversity in terms of video backgrounds and human actions, leading to significant disparities from real-world scenarios. Hence, we collect a video virtual try-on dataset from well-known e-commerce platforms, which contains high-resolution videos of numerous models performing various actions in diverse scenes, together with the corresponding garment image. Moreover, our dataset possesses a greater diversity of garment styles and more intricate patterns on the garments compared to VVT dataset. Visual comparisons between VVT dataset and our dataset are illustrated in the supplementary materials. Our dataset contains $3,156/1,295/7,631$ pairs of human video clips and upper-body garment/lower-body garment/dress images. We randomly split the dataset into training and testing sets according to a ratio of $0.85 : 0.15$ for each garment category.

**Image-based Virtual Try-On.** To further validate the effectiveness of our method, we also conduct experiments on two public image-based virtual try-on datasets (*i.e.*, VITON-HD [5] and Dress Code [29]) using our first-stage model. The VITON-HD dataset contains $13,679$ pairs of upper-body model and garment images, with $2,032$ pairs utilized for testing. The Dress Code dataset contains $15,363/8,951/2,947$ pairs of full-body model and upper-body garment/lower-body garment/dress images, with $1,800$ pairs of each category utilized for testing.

### 4.2 Implementation Details

We initialize the main U-Net with pre-trained Stable Diffusion 2 inpainting[1] model and initialize the garment encoder with Stable

---

[1] https://huggingface.co/stabilityai/stable-diffusion-2-inpainting

Diffusion v2.1 [32]. The pose encoder consists of three convolutional blocks, which down-sample the input DensePose to match the latent's spatial shape. The weights of its final layer are initialized to zero following previous works [20]. We apply data augmentation with random horizontal flipping at a probability of 0.5 to enhance the robustness of the model. For image-based virtual try-on task, our model is trained at a resolution of $512 \times 384$ and $1024 \times 768$ separately, while for video virtual try-on task, we only train at $512 \times 384$ resolution. We adopt Adam [24] optimizer with a constant learning rate of $5 \times 10^{-5}$. All the experiments are conducted using 16 NVIDIA A100 GPUs. In the first stage, we utilize a batch size of 64 for high-resolution and 256 for low-resolution models. In the second stage, we sample 16-frame sequences and set batch size to 64. The model is trained for $20,000$ iterations during the first stage and $10,000$ iterations during the second stage. The weight of perceptual loss $\lambda$ is set to $10^{-3}$. During inference, we set the guidance scale $s_g = 1.5$. We utilize the DDIM [36] sampler, sampling on a single NVIDIA A100 GPU for 50 steps.

### 4.3 Quantitative Results

**Video Virtual Try-On.** We evaluate the performance of video virtual try-on in two test settings. In the paired setting, the model is provided with both a person video and their original garment for reconstruction, while the unpaired setting involves substituting the garment of a person video with a different garment. Following previous work [21], for the paired setting, we utilize two image-level metrics, SSIM [40] and LPIPS [50], to measure the model's ability of reconstructing original videos. For the unpaired setting, we employ the Video Fréchet Inception Distance (VFID) by replacing the 2D image backbone in FID [16] with a 3D backbone I3D [3] to measure the realism and fidelity of generated videos.

Table 1 presents the quantitative results of video virtual try-on task on VVT [8] and our dataset. To validate the effectiveness of our models of both two stages, and highlight the limitations of image-based try-on methods in video try-on task, we compare our method with both image-based virtual try-on and video virtual try-on methods. Our first-stage model surpasses the other two image-based VTON methods and even outperforms video virtual try-on methods, demonstrating its capability to generate high-quality single-frame images. After training in the second stage, our final model surpasses all other methods in terms of all metrics on both datasets. Compared

**Table 2: Quantitative results of image-based virtual try-on task on the VITON-HD dataset.**

| Method | SSIM ↑ | LPIPS ↓ | FID ↓ | KID ↓ |
|---|---|---|---|---|
| HR-VITON [26] | 0.876 | 0.096 | 12.31 | 3.81 |
| GP-VTON [42] | 0.890 | 0.085 | 9.82 | 1.42 |
| LaDI-VTON [28] | 0.875 | 0.091 | 9.32 | 1.55 |
| DCI-VTON [12] | 0.890 | 0.072 | 8.77 | 0.89 |
| StableVITON [23] | 0.878 | 0.075 | 9.43 | 1.54 |
| **GPD-VVTO† (Ours)** | **0.891** | **0.070** | **8.57** | **0.78** |

with the first-stage model, there is a noticeable improvement in the VFID score (−0.51 on VVT and −0.84 on our dataset), demonstrating the significance of temporal attention for video virtual try-on task. Furthermore, our final model exhibits a higher SSIM score compared to the first-stage model, indicating that the model's ability to preserve garment details is improved by referencing garment information in the GTA module.

**Image-based Virtual Try-On.** To further demonstrate the effectiveness of our proposed method, we also evaluate our first-stage model trained at a resolution of $512 \times 384$ on image-based virtual try-on task. Similar to video virtual try-on, we evaluate the performance of image-based virtual try-on under the paired and unpaired settings. We utilize SSIM [40] and LPIPS [50] metrics for the paired setting, and FID [16] and KID [1] for the unpaired setting.

We compare with both GAN-based and diffusion-based methods. Table 2 displays the qualitative results on the VITON-HD [15] dataset. Our proposed GPD-VVTO achieves the best performance across all metrics, demonstrating that our method can generate realistic and natural images while preserving the structure of original image. We notice that the FID score of diffusion-based methods is generally lower than that of GAN-based methods, indicating that diffusion models can generate images with more fidelity. We also conduct experiments on the Dress Code [29] dataset, which contains a greater variety of garment types and presents more challenging scenarios. The results are shown in Table 3. By adjusting the mask strategy, our method can be applied to garments of different categories. Our method outperforms previous methods by a large margin in all the garment categories as well as overall evaluation (−2.57 FID score and −1.88 KID score compared with the second best result), showcasing its capability to handle complex cases.

## 4.4 Qualitative Results

**Video Virtual Try-On.** The left panel of Figure 1 illustrates the video virtual try-on results of our method on the VVT dataset and our collected dataset. The videos in the VVT dataset primarily feature a model walking towards and away from the camera. Our generated results maintain both the fidelity to the target garment and the consistency between frames as the model walks and turns. In our dataset, the video backgrounds and human poses are more complex. Despite this, our generated results still manage to maintain consistency in both background and garment, demonstrating the robustness of our method. More video results are available in the supplementary materials.

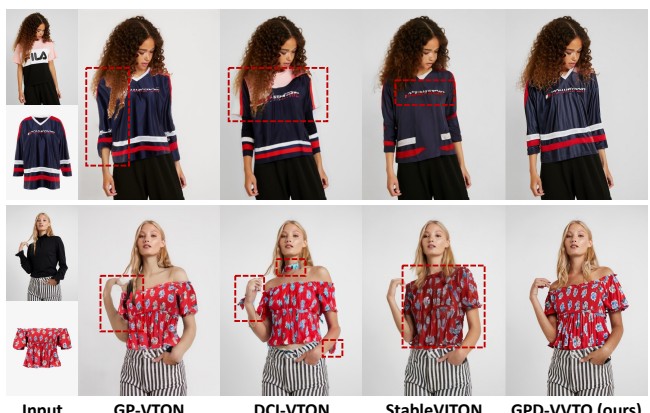

Input          GP-VTON          DCI-VTON          StableVITON          GPD-VVTO (ours)

**Figure 3: Qualitative comparison on VITON-HD dataset. Shortcomings of previous methods are highlighted in red dashed boxes. Better viewed on a zoomed, color monitor.**

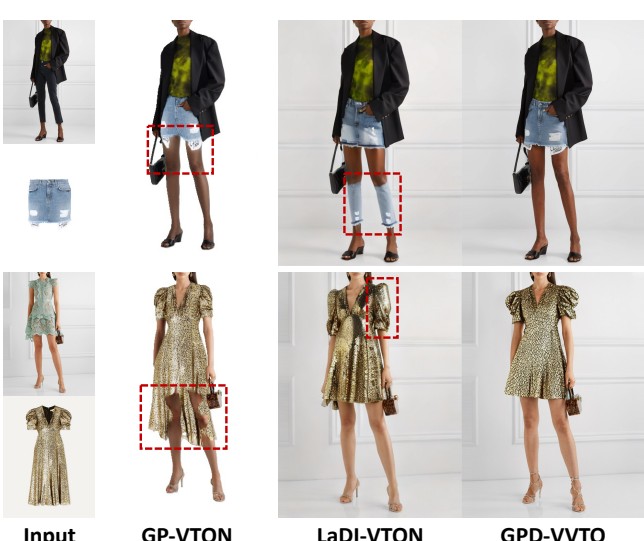

Input          GP-VTON          LaDI-VTON          GPD-VVTO

**Figure 4: Qualitative comparison on Dress Code dataset. Shortcomings of previous methods are highlighted in red dashed boxes. Better viewed on a zoomed, color monitor.**

**Image-based Virtual Try-On.** The right panel of Figure 1 illustrates the image-based virtual try-on results of our method on VITON-HD and Dress Code datasets. Our approach is capable of preserving highly intricate patterns and text on garments, and demonstrates strong robustness in challenging human pose scenarios. We also conduct qualitative comparisons between previous and our method. The qualitative comparison on VITON-HD dataset is shown in Figure 3. In the first row, GP-VTON [42] and DCI-VTON [12] employ separate warping modules, resulting in unnatural blending between the garment and the person. StableVITON [23] achieves a natural fitting, but fails to preserve the text on the garment. In contrast, our method ensures the seamless blending between the garment and the person while retaining the text on the

Table 3: Quantitative results of image-based virtual try-on task on Dress Code dataset. * means the results are reported in previous works.

| Method | Upper-body | | Lower-body | | Dresses | | All | | | |
|---|---|---|---|---|---|---|---|---|---|---|
| | FID ↓ | KID ↓ | FID ↓ | KID ↓ | FID ↓ | KID ↓ | SSIM ↑ | LPIPS ↓ | FID ↓ | KID ↓ |
| PSAD* [29] | 17.51 | 7.15 | 19.68 | 8.90 | 17.07 | 6.66 | 0.918 | 0.058 | 10.61 | 6.17 |
| GP-VTON [42] | 14.80 | 3.22 | 14.05 | 2.60 | 13.76 | 2.55 | 0.918 | 0.069 | 6.98 | 3.02 |
| LaDI-VTON [28] | 14.05 | 2.95 | 14.29 | 2.56 | 14.13 | 3.00 | 0.902 | 0.071 | 6.75 | 2.27 |
| GPD-VVTO† (Ours) | **10.11** | **0.28** | **11.02** | **0.69** | **10.46** | **0.70** | **0.924** | **0.045** | **4.18** | **0.39** |

Table 4: Ablation study to verify the effectiveness of different components of GPD-VVTO.

| Method | SSIM ↑ | LPIPS ↓ | VFID ↓ |
|---|---|---|---|
| w/o SCA modules | 0.925 | 0.062 | 1.31 |
| w/o JSA modules | 0.890 | 0.087 | 1.60 |
| w/o GTA modules | 0.921 | 0.058 | 1.28 |
| **GPD-VVTO** | **0.928** | **0.056** | **1.28** |

Table 5: Effects of the guidance scale $s_g$ and the weight of perceptual loss $\lambda$.

(a) Effects of $s_g$.

| $s_g$ | SSIM ↑ | LPIPS ↓ | VFID ↓ |
|---|---|---|---|
| 1.0 | 0.915 | 0.068 | 1.35 |
| **1.5** | **0.928** | 0.056 | **1.28** |
| 2.0 | 0.926 | **0.055** | 1.30 |
| 2.5 | 0.921 | 0.062 | 1.32 |

(b) Effects of $\lambda$.

| $\lambda$ | SSIM ↑ | LPIPS ↓ | FID ↓ |
|---|---|---|---|
| 0 | 0.920 | 0.062 | 1.31 |
| $10^{-4}$ | 0.921 | 0.060 | 1.31 |
| $10^{-3}$ | **0.928** | **0.056** | **1.28** |
| $10^{-2}$ | 0.923 | 0.060 | 1.30 |

garment. Moreover, our method restores the fabric of the garment more faithfully. In the second row, the target garment is an off-shoulder top. While GP-VTON and DCI-VTON manage to preserve the appearance of the garment, they exhibit artifacts around the arms and neck. StableVITON changes the style of the target garment and introduces color discrepancies. In contrast, our method preserves the style of the garment while retaining its appearance. Similar phenomenons can be observed in the qualitative comparison on Dress Code dataset as shown in Figure 4. GP-VTON [42] and LaDI-VTON [28] fail to preserve the local texture or overall style of the target garment and exhibit artifacts in the blending of the person and the garment. On the contrary, our method excels in both the naturalness and fidelity of the garment.

## 4.5 Ablation Studies

**Different components of GPD-VVTO.** The quantitative results of ablating different components in GPD-VVTO on the VVT dataset are reported in Table 4. In the 1-st row, we replace the garment features used in the cross-attention modules with the final output of the DINO encoder instead of multiple intermediate features. The performance exhibits some decline compared with our full model, indicating that the SCA modules enhance the quality of the generated results by leveraging semantic features from different abstraction levels. In the 2-nd row, we discard the garment encoder and substitute the JSA modules with standard self-attention. We observe a significant decrease in the model's performance, demonstrating the crucial impact of multi-scale dense garment features on the try-on results. In the 3-rd row, we replace the GTA modules with normal temporal attention without referring to garment features. The LPIPS and VFID scores remain relatively consistent with the full model, but there is a notable discrepancy in SSIM, indicating that concatenating garment feature sequences in the GTA module benefits the preservation of garment information. Overall, the modules proposed in our method enhance the fidelity to the garment and temporal consistency of the generated results.

**Effects of hyperparameters.** We also investigate the effects of hyperparameters on the performance of our model, including the guidance scale $s_g$ and the weight of perceptual loss $\lambda$. Table 5 (a) presents the ablation study on $s_g$. When $s_g = 1$, the classifier-free guidance is not activated, resulting in relatively inferior performance. As $s_g$ increases, the performance of the model first improves and then declines. We find that a larger guidance scale can preserve more complete patterns, but there is a significant color discrepancy of the garment. Taking into account the overall performance of the model under different values of $s_g$, we set $s_g = 1.5$ according to experimental results. Table 5 (b) shows the effects of changing the value of $\lambda$. As $\lambda$ becomes larger, the weight of the perceptual loss gradually increases. We observe that the model's performance first improves and then declines. This may be because when $\lambda$ is small, the perceptual loss has little effect; and when $\lambda$ is large, the update of model's parameters relies more on the perceptual loss rather than the MSE loss, leading to a degradation of the quality of generated results. Given the ablation results, we set $\lambda = 10^{-3}$ in our experiments.

## 5 CONCLUSION

In this paper, we focus on the task of video virtual try-on and propose a LDM-based two-stage framework for the first time. Our method leverages both global semantic features and local texture features of the garment to preserve garment details as much as possible. We devise a novel Garment-aware Temporal Attention module that integrates features of the garment into temporal attention, improving the temporal consistency of the generated videos while maintaining the fidelity to the garment. Furthermore, we collect an in-the-wild video virtual try-on dataset to address the limited variety in terms of video background and human actions of current datasets. Extensive experimental results demonstrate that our method outperforms previous state-of-the-art methods on both image-based and video-based virtual try-on datasets, indicating the effectiveness of our proposed framework.

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
