# OpenReview forum: "GPD-VVTO: Preserving Garment Details in Video Virtual Try-On"
_acmmm.org/ACMMM/2024/Conference — MM2024 Poster_

### Official Review · Reviewer_DVz9 · 2024-05-09

**Rating:** 4
**Confidence:** 3

**Summary:**

The paper focuses on enhancing the realism and consistency of virtual try-on technologies for videos using a novel two-stage framework based on Latent Diffusion Models (LDMs). The main goal is to accurately transfer garments onto a person in a video, maintaining high-quality single-frame results and ensuring temporal consistency throughout the video sequence. The paper involves a two-stage approach. The first stage focuses on generating high-quality images for individual frames using a diffusion model that integrates both low-level texture and high-level semantic features of garments. The second stage enhances the model with video data to maintain temporal consistency using a novel Garment-aware Temporal Attention (GTA) mechanism. The paper also highlights the creation of a new video virtual try-on dataset that features diverse backgrounds and human actions to better reflect real-world applications. Extensive experiments show that this method outperforms existing state-of-the-art techniques in both image-based and video-based virtual try-on tasks.

**Strengths:**

1. The approach introduces a two-stage framework utilizing Latent Diffusion Models (LDMs) specifically tailored for video virtual try-on applications. This is a novel adaptation as previous works predominantly focus on static image-based virtual try-on.
2. The introduction of the Garment-aware Temporal Attention (GTA) module is particularly innovative, addressing the challenge of maintaining garment fidelity throughout the motion in videos, a feature not extensively explored in prior research.
3. The video try-on data set proposed in this paper makes up for the lack of data in this field.

**Limitations:**

1.	The author's representation of the attention mechanism using Attn() in equations (3) to (5) is not sufficiently formalized. It is advisable to either explicitly detail the attention calculation formula or define Attn() precisely, e.g., Attn(Q,K,V) = ...
2.	The manuscript omits a discussion on the methodological limitations. For robustness, performance analysis under atypical scenarios, such as individuals wearing unconventional clothing, should be included.
3.	The performance evaluation of the algorithm relies on LPIPS and VFID metrics to assess temporal consistency, which are primarily designed to evaluate consistency in video frame quality. The paper lacks an assessment of motion coherency. Given the assertion in line 152 regarding the consistency and coherence of actions and garment appearance throughout the video, incorporating a metric for evaluating coherence, such as dtSSD[1], is recommended.
4.	The documentation concerning the dataset used is inadequate. The authors need to provide additional details about the data collection and preprocessing procedures, selection criteria, data filtering, handling of personally identifiable information, ethical considerations, dataset accessibility (public or private), and other pertinent information.

[1] M. Erofeev, Yury Gitman, D. Vatolin, Alexey Fedorov, and J. Wang. Perceptually motivated benchmark for video matting. In BMVC, 2015.

**Suitability:**

3

---

### Official Review · Reviewer_diwH · 2024-05-17

**Rating:** 3
**Confidence:** 4

**Summary:**

This paper focuses on improving video virtual try-on (VVT) technologies, where a garment from a still image is realistically animated onto a person in a video. The authors introduce a two-stage Latent Diffusion Model framework called Garment-Preserving Diffusion for Video Virtual Try-On (GPD-VVTO). This method first trains on single-frame images to ensure high-quality garment visualization while preserving details. It then extends to video data to enhance temporal consistency using a Garment-aware Temporal Attention module, which integrates garment features into the temporal modeling process.

**Strengths:**

1.  The final experimental outcomes for GPD-VVTO demonstrate robust performance in both Image-based and Video Virtual Try-On comparisons.
2.  The GTA Module can improve the temporal consistency of the generated video and is corroborated by experimental results.
3.  This paper is well-organized and easy to follow.

**Limitations:**

1.  In terms of novelty, IDM-VTON[1] have similarities with the SCA and JSA modules, which somewhat limit the novelty of this paper.
2.  Apart from the GTA module, the main differences between this paper and IDM-VTON lie in the use of the DINO Encoder (as opposed to IDM-VTON's IP-Adapter) and the addition of a Pose Encoder. Conducting ablation experiments on these modules will help verify their effectiveness.
3.  Table 3 lacks SSIM and LPIPS metrics for the three subcategories of the Dresscode dataset.
4.  The video virtual try-on dataset is one of the key contributions of this paper. Therefore, I would like to ask if there are plans to release the code and dataset, and if so, do you have a tentative date in mind?

[1] Choi Y, Kwak S, Lee K, et al. Improving Diffusion Models for Virtual Try-on[J]. arXiv preprint arXiv:2403.05139, 2024.

**Suitability:**

3

---

### Official Review · Reviewer_2wh5 · 2024-05-24

**Rating:** 3
**Confidence:** 4

**Summary:**

With the rise of e-commerce, video virtual try-on has garnered widespread attention in the industry, offering consumers an immersive and interactive online shopping experience. This work introduces an LDM-based two-stage framework, GPD-VVTO, to ensure realistic and coherent garment transfers in videos. By integrating garment features into the denoising network, this method preserves garment details and maintains temporal consistency throughout the video. Additionally, to address the lack of diversity in existing datasets, this method collects a video virtual try-on dataset from e-commerce platforms, containing high-resolution videos depicting diverse scenes and human actions, which is more beneficial for real-world applications. The proposed method outperforms state-of-the-art approaches in both image-based and video-based virtual try-on tasks.

**Strengths:**

-1. This work explores the capability of diffusion models in video virtual try-on, and proposes a two-stage framework that leverages both global semantic features and local texture features of the garment to preserve garment details.

-2. This work devises a novel Garment-aware Temporal Attention module that integrates features of the garment into temporal attention, improving the temporal consistency of the generated videos while maintaining fidelity to the garment.

-3. This work collects a video virtual try-on dataset with more diversity to resolve the limited variety of current datasets in terms of video background and human actions.

**Limitations:**

-1. From Figure 2, it appears that, apart from the JSA and GTA modules, this architecture is almost identical to  IDM-VTON [1] and OOTDiffusion [2]. What are the fundamental differences in principle?

[1] Improving Diffusion Models for Virtual Try-on.

[2] OOTDiffusion: Outfitting Fusion based Latent Diffusion for Controllable Virtual Try-on.

-2. This paper uses existing methods such as [10][47][13] to preprocess the training samples. If these methods ([10][47][13]) generate unreliable results, would they negatively impact the outcomes? How should this be addressed?

-3. If $m$ incorrectly segments $x$, the resulting output will lose identity information. As shown in Figure 3, both GP-VTON and DCI-VTON faithfully preserve the shape of the hair, whereas the method in this paper is clearly at a disadvantage in this regard.

-4. How is perceptual loss $L_{𝑝𝑒𝑟𝑐𝑒𝑝𝑡𝑢𝑎l}$ applied? Please explain in detail.

-5. Is there a plan to open-source the collected dataset? If not, how can the effectiveness of this method be fairly validated?

-6. There is little information about the collected dataset. Please add an analysis.

-7. In Table 2, the values for the StableVITON method seem to differ from those in the original paper. Why does this discrepancy exist?

-8. Please add more baseline methods to the quantitative results.

-9. Please add continuous comparison results reflecting video virtual try-on in the main text, not just single-frame results.

-10. From Table 4, it can be seen that the performance w/o GTA is almost identical to the performance w/ GTA, indicating that GTA does not provide significant performance gains.

-11. Providing visualized ablation experiment results can help the authors intuitively understand the contributions of each module.

**Suitability:**

2

---

### Official Review · Reviewer_Sogi · 2024-05-24

**Rating:** 3
**Confidence:** 3

**Summary:**

This paper presents a new method for video virtual try-on called Garment-Preserving Diffusion for Video Virtual Try-On (GPD-VVTO). The method tackles two key issues in Video Virtual Try-On by a two-stage framework based on a latent diffusion model: the realistic naturalness of the single-frame results and the consistency of the character's movements with the clothing throughout the video. In the first stage, the model is trained on single-frame data to generate high-quality fitting images and maintains clothing details by integrating low-level texture features and high-level semantic features. In the second phase, the model is trained on video data to enhance temporal consistency, and a new garment-aware temporal attention module is introduced to maintain garment coherence. In addition, the authors collected a high-resolution video virtual fitting dataset containing multiple scenarios. Experimental results show that the method outperforms existing state-of-the-art methods in both image and video virtual fitting tasks.

**Strengths:**

1. the paper is well expressed and can help the reader to understand it better.
2. the author clearly describes the methodology designed in the paper and the configuration of the experiment is described in great detail.
3. the ablation study of the paper is detailed and the key components mentioned in the paper are ablated. The work is experimentally adequate.

**Limitations:**

1. The paper aims to accomplish the Video Virtual Try-On task, but lacks a comparative analysis of the processing frame rate with the baseline method.
2. What are the significant performance and structural differences of the proposed method compared to IDM-VTON (Improving Diffusion Models for Virtual Try-on)? What are the different design concepts?
3. lack of comparison with more baseline methods on Video Virtual Try-On.
4. please analyse specifically the dataset you have collected.
5. Table 1 is missing key quantitative data for Video VTON on its own dataset.
6. the paper aims to solve the Video Virtual Try-On task, but does not show the crucial visualisation results in the qualitative experimental part. Instead, the results presented in Figure 1 lack a comparison with the baseline.
7. The visualisation results corresponding to Table 4 are missing in the ablation experiments.

**Suitability:**

2

---

### Meta-Review · Area_Chair_HDqp · 2024-07-07

**Recommendation:** Accept (Poster)
**Confidence:** 4

**Metareview:**

The paper proposes a Latent Diffusion-based method for virtual try-ons for videos.  The goal is to accurately and realistically transfer garments onto a person in a video.  The method is based on a two-stage approach, first focusing on a single-frame image to ensure high quality and then extending it to video data to maintain temporal consistency.
The method is scientifically sound, effectively exploring the capabilities of diffusion models for virtual try-on.  The experimental setup is well-designed, and the results and ablation studies show that the method outperforms previous SOTA methods (including a concurrent one, IDM-VTON).  Moreover, the authors plan to make the dataset available, which can, which can significantly aid the community in further exploring this important topic.  Some of the reviewers raised concerns, in particular, in terms of novelty compared to the concurrent work IDM-VTON.  During the rebuttal, the authors provided more information on the dataset collection, further compared its method with IDM-VTON, and provided additional quantitative results, showing that their method outperforms additional baselines.

Given the importance of the work, the perfect fit for the conference, and the technical contribution together with the release of the dataset, the recommendation is to 'accept' the paper.